# Breaking through the Mermin-Wagner limit in 2D van der Waals magnets

Sarah Jenkins[1,2,3], Levente Rózsa [4], Unai Atxitia[5,6], Richard F. L. Evans [1], Kostya S. Novoselov [7] & Elton J. G. Santos [8,9,10] ✉

The Mermin-Wagner theorem states that long-range magnetic order does not exist in one- (1D) or two-dimensional (2D) isotropic magnets with short-ranged interactions. Here we show that in finite-size 2D van der Waals magnets typically found in lab setups (within millimetres), short-range interactions can be large enough to allow the stabilisation of magnetic order at finite temperatures without any magnetic anisotropy. We demonstrate that magnetic ordering can be created in 2D flakes independent of the lattice symmetry due to the intrinsic nature of the spin exchange interactions and finite-size effects. Surprisingly we find that the crossover temperature, where the intrinsic magnetisation changes from superparamagnetic to a completely disordered paramagnetic regime, is weakly dependent on the system length, requiring giant sizes (*e.g.*, of the order of the observable universe ~ $10^{26}$ m) to observe the vanishing of the magnetic order as expected from the Mermin-Wagner theorem. Our findings indicate exchange interactions as the main ingredient for 2D magnetism.

The demand for computational power is increasing exponentially, following the amount of data generated across different devices, applications and cloud platforms[1,2]. To keep up with this trend, smaller and increasingly energy-efficient devices must be developed, which require the study of compounds not yet explored in data-storage technologies. The discovery of magnetically stable 2D vdW materials could allow for the development of spintronic devices with unprecedented power efficiency and computing capabilities that would, in principle, address some of these challenges[3]. Indeed, the magnetic stability of vdW layers has been one of the central limitations for finding suitable candidates, given that strong thermal fluctuations are able to rule out any magnetism. As it was initially pointed out by Hohenberg[4] for a superfluid or a superconductor, and extended by Mermin and Wagner[5] for spins on a lattice, long-range order should be suppressed at finite temperatures in the 2D regime, when only short-range isotropic interactions exist. Importantly, the theorem only

excludes long-range magnetic order at finite temperature in the thermodynamic limit[5], i.e., for infinite system sizes. However, the common understanding is that the theorem also excludes the alignment of spins in samples studied experimentally which are a few micrometres in size[6,7], suggesting that such systems are indistinguishable from infinite. Previous reports[8–17] have discussed at different levels of theoretical and experimental approaches the limitations and the potential ways to overcome the Mermin-Wagner theorem, which provides a historical evolution of the common concepts used in the field of 2D magnetism.

The long-range order characterising infinite systems only becomes distinguishable from short-range order describing the local alignment of the spins if the system size exceeds the correlation length at a given temperature[18]. Previous numerical studies and the scaling analysis of 2D Heisenberg magnets[19–22] have established that although only short-range order is observable at finite temperature, the spin correlation length can be larger than the system size below some finite crossover temperature.

[1]Department of Physics, University of York, York YO10 5DD, UK. [2]TWIST Group, Institut für Physik, Johannes Gutenberg Universität, 55128 Mainz, Germany. [3]TWIST Group, Institut für Physik, Universität Duisburg-Essen, Campus Duisburg 47057 Duisburg, Germany. [4]Fachbereich Physik, Universität Konstanz, D-78457 Konstanz, Germany. [5]Dahlem Center for Complex Quantum Systems and Fachbereich Physik, Freie Universität Berlin, 14195 Berlin, Germany. [6]Instituto de Ciencia de Materiales de Madrid, CSIC, Cantoblanco 28049 Madrid, Spain. [7]Institute for Functional Intelligent Materials, National University of Singapore, Singapore 117544, Singapore. [8]Institute for Condensed Matter Physics and Complex Systems, School of Physics and Astronomy, The University of Edinburgh, Edinburgh EH9 3FD, United Kingdom. [9]Donostia International Physics Center (DIPC), 20018 Donostia-San Sebastián, Basque Country, Spain. [10]Higgs Centre for Theoretical Physics, The University of Edinburgh, Edinburgh EH9 3FD, United Kingdom. ✉e-mail: esantos@ed.ac.uk

An intriguing question on this long-range limit is how we can understand real-life materials, which routinely have a finite size $L$ (Fig. 1a), in light of the Mermin-Wagner theorem. It is known that thermal fluctuations will affect the emergence of spontaneous magnetisation at low dimensionality. Nevertheless, it is unclear which kind of spin ordering can be foreseen in thin vdW layered compounds when finite-size effects and exchange interactions play together. With recent advances in computational power and parallelisation scalability, it is possible to directly model magnetic ordering processes and dynamics of 2D materials on the micrometre length-scale accessible experimentally.

Here, we show that short-range order can exist in systems with no anisotropy, even down to the 1D and 2D limits. By using computer-intensive atomistic spin simulations and analytical models, we demonstrate the non-applicability of the Mermin–Wagner theorem for practical length scales and device implementations. The theorem requires that the thermodynamic limit be taken and only for distances beyond the diameter of the observable universe, as revealed by our results, it might be valid. The large distance character of short-range interactions in 2D vdW magnets drives the formation of magnetic ordering at different lattice symmetries, flakes shapes and chemical compositions. Our results unveil that exchange interactions are the main driving force behind the stabilisation of 2D magnetism and broaden the horizons of possibilities for the exploration of compounds with low anisotropy at an atomically thin level.

## Results

We start by defining the magnetisation in our systems as:

$$\mathbf{m} = \frac{1}{N} \sum_i \mathbf{S}_i, \tag{1}$$

where $\mathbf{S}_i$ denotes the classical spin unit vector at lattice site $i$ and $N$ is the number of sites. In the absence of external magnetic fields, the expectation value of the magnetisation $\langle \mathbf{m} \rangle$ vanishes in any finite-size system due to time-reversal invariance. Yet, 3D systems of only a few nanometres in size that are far from infinite have been studied for decades and exhibit a clear crossover from a magnetically ordered to a paramagnetic phase[23,24]. The Mermin-Wagner theorem establishes that $\langle \mathbf{m} \rangle$ must also be zero in infinite 2D systems with short-ranged isotropic interactions. However, for practical implementations it is relevant to unveil whether the average magnetisation vanishes because the spins are completely disordered at any point in time, or if they are still aligned on short distances but the overall direction of the magnetisation $\mathbf{m}$ strongly suffers time-dependent variation. Short-range order may be characterised by the intrinsic magnetisation[25]:

$$\langle |\mathbf{m}| \rangle = \left\langle \sqrt{\left( \frac{1}{N} \sum_i \mathbf{S}_i \right)^2} \right\rangle, \tag{2}$$

which is always positive by definition. The intrinsic magnetisation is $\langle |\mathbf{m}| \rangle \approx 1$ in the short-range-ordered regime and converges to zero when the spins become completely disordered[6,26,27].

For simplicity we first consider a 2D honeycomb lattice (Fig. 1a) to model the magnetic ordering process for a large flake of $1000 \times 1000$ nm$^2$. Such a symmetry is very common in several vdW materials holding magnetic properties and interfaces[3,28], such as $Cr_2Ge_2Te_6$ (CGT) or $CrI_3$ in which 2D magnetic ordering was first discovered[29,30]. The system consists of 8 million atoms with nearest-neighbour Heisenberg exchange interactions $J_{ij}$ and no magnetic anisotropy ($K$) described via highly accurate Monte Carlo simulations (see

**Fig. 1 | Short-range magnetic ordering at finite temperatures in a 2D isotropic magnet. a** Local view of the spin directions extracted from the atomistic simulations on a 2D honeycomb lattice. $a$ is the atomic spacing ($a = 0.4$ nm), L is the length considered in the computations, and $\mathbf{M}_{av}$ is the averaged magnetisation vector. $\theta$ corresponds to the angle between $\mathbf{M}_{av}$ and the $z$-axis. $\theta_0 = 0$ denotes the initial configuration aligned with the $z$-axis. **b** Temperature-dependent intrinsic magnetisation $\langle |\mathbf{m}| \rangle$ with ($K = 1 \times 10^{-24}$ J/atom) and without ($K = 0$) anisotropy in a $1000 \times 1000$ nm$^2$ flake. Solid lines are the fit to Eq. (3). For $K = 0$, the fitting parameters are $\beta = 0.54 \pm 0.020$ and $T_x = 23.342 \pm 0.237$ K. For $K > 0$, $\beta = 0.427 \pm 0.021$ and $T_x = 26.543 \pm 0.320$ K. **c, d** Temporal variation of the magnetisation ($m/m_s$) and angle $\theta - \theta_0$, respectively, at $T = 10$ K. All three spatial components ($x, y, z$) are considered in **c**. The dashed line in **d** shows the initial state in the simulations.

Supplementary Sections 1–2 for details). We use an isotropic Heisenberg spin Hamiltonian $\mathcal{H} = -\sum_{i<j} J_{ij}\mathbf{S}_i \cdot \mathbf{S}_j$ as stated in the Mermin–Wagner theorem[5]. As it is shown below, our conclusions do not depend on the magnitude of the exchange interactions chosen. Nevertheless, to give a flavour of a potential material to study, we set $J_{ij}$ to similar values to those obtained for CGT layers[29] where a negligible magnetic anisotropy (<1 μeV) was observed for thin layers but yet a stable magnetic signal was measured at finite temperatures (~4.7 K). We begin by assessing the existence of any magnetic order at non-zero temperatures by equilibrating the system for $39 \times 10^6$ Monte Carlo steps using a uniform sampling[31] to avoid any potential bias before a final averaging at thermal equilibrium for a further $10^6$ Monte Carlo steps.

Strikingly, a crossover between the low-temperature short-range-ordered regime and the completely disordered state ($\langle|\mathbf{m}|\rangle \approx 0$) is observed at nonzero temperatures (Fig. 1b) and zero magnetic anisotropy ($K = 0$). To estimate the crossover temperature ($T_x$), the simulation data was fitted by the Curie–Bloch equation in the classical limit[6]:

$$\langle|\mathbf{m}|\rangle(T) = \left(1 - \frac{T}{T_x}\right)^\beta,\qquad(3)$$

where $T$ is the temperature and $\beta$ is an exponent in the fitting. From the fitting one obtains $T_x = 23.342 \pm 0.237$ K ($\beta = 0.54 \pm 0.020$), which is about one-third of the mean-field (MF) critical temperature $T_c^{\mathrm{MF}} = zJ_{ij}/(3k_B) = 70.8$ K (where $z = 3$ is the number of nearest neighbours) even for this considerable system size. The simulations were then repeated, including magnetic anisotropy ($K = 1 \times 10^{-24}$ J/atom), which resulted in a slight increase in the crossover temperature ($T_x = 26.543 \pm 0.320$ K, $\beta = 0.427 \pm 0.021$) (Fig. 1b). We observed that this difference in $T_x$ between isotropic and anisotropic cases becomes negligible as the flake size is reduced ($100 \times 100$ nm$^2$) with minor variations of the curvature of the magnetisation versus temperature (Supplementary Section 3 and Supplementary Fig. 1). We also checked that different Monte Carlo sampling algorithms (i.e., adaptive) and starting spin configurations (i.e., ordered, disordered) do not modify the overall conclusions (Supplementary Section 4 and Supplementary Fig. 2). Taking dipolar interactions into account only has a minor effect on the intrinsic magnetisation curve (Supplementary Fig. 3). Although the magnetocrystalline anisotropy $K$ or the dipolar interactions circumvent the Mermin-Wagner theorem and lead to a finite critical temperature, this indicates that systems up to lateral sizes of 1 μm are not suitable for observing the critical behaviour. Instead the crossover in the short-range order defined by the isotropic interactions dominates in this regime, regardless of whether the anisotropy is present or absent. Previous studies on finite magnetic clusters on metallic surfaces[32,33] suggested that anisotropy is not the key factor in the stabilisation of magnetic properties at low dimensionality and finite temperatures, but rather it determines the orientation of the magnetisation.

Even though short-range interactions can stabilise short-range magnetic order in 2D vdW magnetic materials, this does not necessarily imply that the direction or the magnitude of the magnetisation is stable over time. As thermally activated magnetisation dynamics may potentially change spin directions[34], it is important to clarify whether angular variations of the spins are present. Hence we compute the time evolution of the magnetisation along different directions $(x, y, z)$ and its angular dependence (Fig. 1c, d) through the numerical solution of the Landau-Lifshitz-Gilbert equation (see Methods for details). Over the whole simulation (40 ns), all components of the magnetisation assume approximately constant values which deviate by $\pm 5°$ from the mean direction $\theta_{\mathrm{av}}$. Similar analyses undertaken for different flake sizes ($L \times L$, $L = 50, 100, 500$ nm) show that the spin direction is very stable at each temperature considered (2.5 K, 10 K, 20 K, 30 K, 40 K) and follows a Boltzmann distribution

(Supplementary Section 5 and Supplementary Fig. 4). These results show that the magnetisation in a 2D isotropic magnet is not only stable in magnitude but its direction only negligibly varies over time.

An outstanding question raised by the modelling of the 2D finite flakes is whether other kind of common lattice symmetries (i.e., hexagonal, square), lower dimensions (i.e., 1D) and different sizes may follow similar behaviour to that found in the honeycomb lattice. Figure 2 shows that the effect is universal regardless of the details of the lattice or the dimension considered. We find persistent magnetic order for $T > 0$ K at zero magnetic anisotropy for the cases considered. There is a consistent reduction in the crossover temperature as a function of the system size $L \to \infty$ in agreement with the general trend of the temperature dependence of the correlation length discussed above (Fig. 2a–c). The 1D model (atomic chain) displays a similar trend (Fig. 2d) although the variation of $\langle|\mathbf{m}|\rangle$ with $T$ is different due to the lower dimensionality. We have also checked that several additional factors do not affect these conclusions, such as i) the type of boundary conditions, e.g., open; ii) flake shape (e.g., circular), and iii) strength of the exchange interactions. Supplementary Figs. 5 and 6 provide a summary of this analysis. Indeed, the stabilisation of magnetism in 2D is independent of the magnitude of the exchange interactions considered, as a linear re-scaling of the temperatures is obtained for different $J_{ij}$ values. This indicates the generality of the results which are valid regardless of the chemical details of the 2D material and its corresponding $J_{ij}$ interactions. Moreover, if the exchange coupling between atoms could be engineered via chemical synthesis[35–37], then magnets with either low or high crossover temperatures might be fabricated depending on the target application. Such a procedure would not require heavy elements with sizeable spin orbit-coupling for the generation of magnetic anisotropy since it is not necessary for 2D magnetism.

To give an analytical description of these effects, we use the anisotropic spherical model (ASM) for the calculation of the finite-size effects on the intrinsic magnetisation[25,38,39] (see Supplementary Section 6 for details). The ASM takes into account Goldstone modes in the system and self-consistently generates a gap in the correlation functions which avoids infra-red divergences responsible for the absence of long-range order for isotropic systems in dimensions $d \leq 2$ as $L \to \infty$ as per the Mermin-Wagner theorem. We applied the formalism to 1D and 2D systems for the isotropic Heisenberg Hamiltonian in the absence of an external magnetic field[25]. The results of our analytical calculations are shown as shaded regions in Fig. 2 (see Supplementary Section 6 for the definition of the regions). At low temperatures both limits agree well with our Monte Carlo calculations within the statistical noise and clearly show the existence of a finite intrinsic magnetisation at non-zero temperature for finite size. At higher temperatures there is a systematic difference between the degree of magnetic ordering between the simulations and the analytical calculations due to the ASM only becoming exact in the limit of infinitely many spin components. The large number of Monte Carlo steps and strict convergence criteria to the same thermodynamic equilibrium for ordered and disordered starting states (Supplementary Section 4) rule out critical slowing down[40] as a source of difference between the analytical calculations and the simulations.

One may also argue in terms of the correlation length $\xi$ which is comparable to the system size at the crossover temperature. It has been demonstrated[20] that $\xi \propto \exp(cJ/T)$, where $c$ is a constant, meaning that the inverse crossover temperature $T_x^{-1}$ only logarithmically increases with the system size. Although our simulations are at the limit of the capabilities of current supercomputers, this effect is expected to persist for larger sizes of 2–10 μm. These values represent typical sizes of continuous 2D microflakes in experiments, and much larger than the ideal nanoscale devices likely to be used in future 2D spintronic applications. Fitting a scaling function to the crossover temperatures for different lattice symmetries (Fig. 2), we can plot the

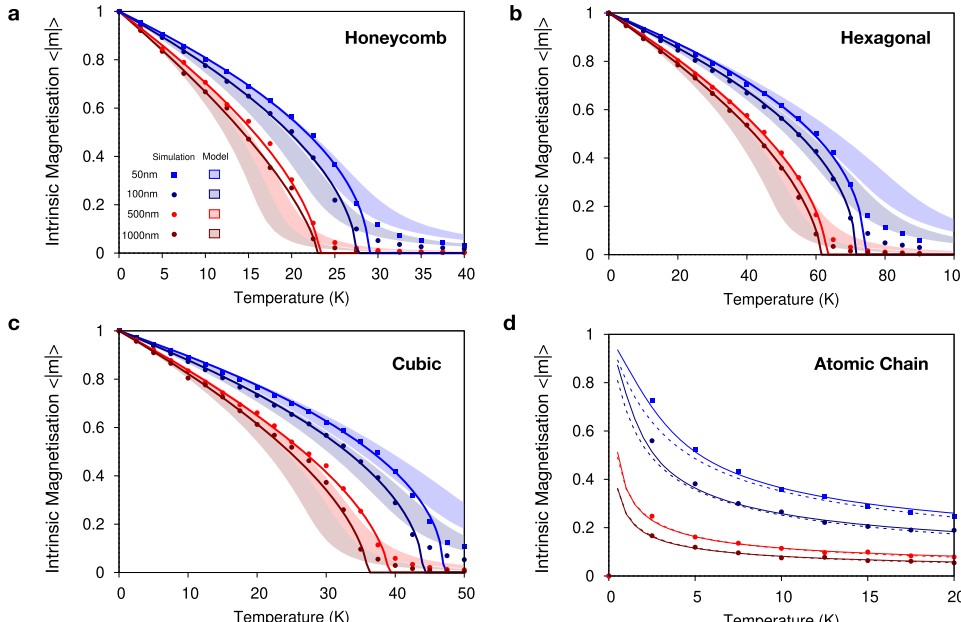

**Fig. 2 | Temperature- and size-dependent properties of isotropic 1D and 2D materials with different crystal structures. a–d** Comparative simulations of the temperature-dependent magnetisation for honeycomb, hexagonal, square lattices and an atomic chain (1D), respectively, for different system sizes. Points indicate the results of Monte Carlo simulations, the lines show fits to the Curie-Bloch Eq. (3) in the classical limit, and the shaded regions indicate the anisotropic spherical model calculations for different assumptions of the renormalisation factor for the Curie temperature arising from the mean-field approximation. See Supplementary Section 6 for details. The dashed and solid lines in **d** indicate the anisotropic spherical model calculations, and the exact solution, respectively. Both show a sound agreement with the atomistic simulations. The datasets in **a**–**c** clearly show the existence of short-range collinear magnetic order for all 2D lattices at the simulated sizes considered with nonzero crossover temperature. Zero magnetic anisotropy is included in all calculations.

scaling of the crossover temperature with size (Fig. 3a), which can then be extrapolated to larger scales. The crossover temperature is still approximately 30 K for 2–10 μm flakes (Fig. 3b). The graph can be extrapolated to show that only at the $10^{15} - 10^{25}$ m range does the crossover temperature become lower than ~1 K. To put these numbers into perspective for physical systems, these length scales lie between the distance of the Earth to the Sun and the diameter of the observable universe. Therefore, the often asserted notion[3] that experimental 2D magnetic samples can be classified as infinite and therefore display no net magnetic order at nonzero temperatures, as expected from the Mermin–Wagner theorem, is not applicable. Surprisingly, simple estimations by Leggett[41] for the stability of graphene crystals following the Mermin–Wagner theorem would require sample sizes of the order of the distance from the Earth to the Moon, which are in sound agreement with our simulation results.

The significance of the crossover temperature $T_x$ in relation to the Curie temperature $T_C$ is particularly important when discussing the nature of the magnetic ordering in 2D magnets at zero anisotropy for $T > 0$ K. We investigate this behaviour through colour maps of the spin ordering after 40 million Monte Carlo steps comparing different system sizes and temperatures (Fig. 4). At very low temperatures $T = 2.5$ K, where there is a high degree of order, the spin directions are highly correlated, as indicated by a mostly uniform colouring. Although the temperatures are near zero, the system is superparamagnetic indicating that over time the magnetisation direction fluctuates, and the effect is most apparent for the smallest sizes where the average direction has moved significantly from the initial direction $\mathbf{S}\|z$. At higher temperatures, the deviation of the spin directions within the sample increases as indicated by the more varied colouring. To quantitatively assess the spin deviations we plot the statistical distribution of angle between the spin direction and the mean direction for different temperatures for each size (Supplementary Fig. 4). For an isotropic distribution on the unit sphere there is a $\sin(\theta)$ weighting, which is seen at the highest temperature for all system sizes. For lower temperatures where the

spin directions are more correlated, the distribution is biased towards lower angles. Qualitatively there is little difference in the spin distributions for the different samples. At $T = 20$ K, there is, however, a systematic trend in the peak angle increasing from $\theta = 40°$ for the $50 \times 50$ nm² flake (Supplementary Fig. 4a) to around $\theta = 60°$ at $1000 \times 1000$ nm² (Supplementary Fig. 4d) indicating an increased level of disorder averaged over the whole sample. This effect is straightforwardly explained by the size dependence of spin-spin correlations (Supplementary Fig. 7). At small sizes the spins are strongly exchange coupled, preventing large local deviations of the spin directions. At longer length scales available for the larger systems, the variations in the magnetisation direction are also larger. Surprisingly, our calculations reveal that this effect is weak: even for very large flakes of a micrometre in size, only a small increase can be observed in the position of the peak in the angle distribution at a fixed temperature. Above the crossover temperature, the spin-spin correlation length becomes very small compared to the system size with rapid local changes in the magnetisation direction, indicative of a completely disordered paramagnetic state. Our analysis reveals that the spins in finite-sized 2D isotropic magnets are strongly aligned due to short-range order at nonzero temperatures and up to the crossover temperature.

## Discussion

Mathematically a phase transition is defined as a non-analytic change in the state variable for the system, such as the particle density or the magnetisation in the case of spin systems. For any finite system the state variable is continuous by definition due to a finite number of particles, forming a continuous path of intermediate states between two distinct physical phases[42]. The same is true for a magnetic system, forming a continuous path between an ordered and a paramagnetic state. A priori then, it is impossible to have a true phase transition for any finite magnetic samples which are routinely implemented in device platforms. Yet, nanoscale magnets that are far from infinite have been studied for decades and exhibit a clear crossover from magnetically

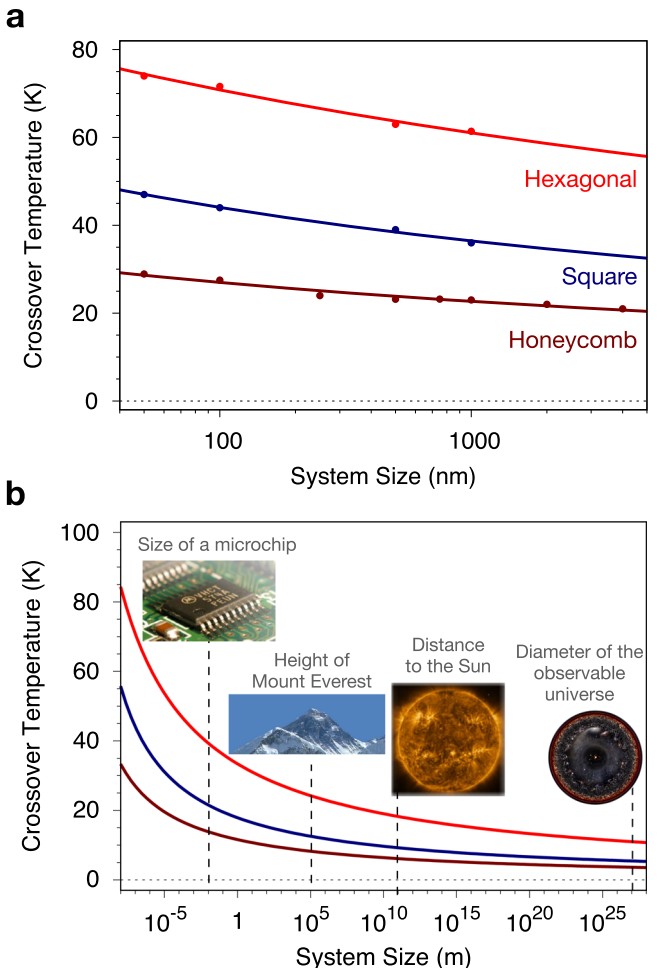

**a**

**b**

**Fig. 3 | Size scaling of the simulated crossover temperature for the different 2D lattices. a** Variation of the crossover temperature $T_x$ with system size for different symmetries (Hexagonal, Square, Honeycomb) on a log-scale. The curves are a fit using $T_x = A/\log(L/B)$, where $A$ and $B$ are fitting constants and $L$ is the system size. $A$ and $B$ are 327.28 K and 0.000542 nm, 484.96 K and 0.00166 nm and 1018.50 K and $5.7 \times 10^{-5}$ nm for honeycomb, square and hexagonal lattices, respectively.
**b** Extrapolation of the exponential fits in **a** to larger sizes on all studied symmetries. The crossover temperature remains finite (>4 K) for systems as large as ~ $10^{25}$ m indicating no dependence of the magnetic anisotropy for stabilisation of magnetic ordering. Insets provide a comparison with physical distances observed in different systems. Figures in **b** are adapted with permission under a Creative Commons CC BY license from Wiki Commons. Microchip: Integrated circuit on a microchip by Jon Sullivan, 2006, at Public Domain from Wiki Commons. Sun: inset is from ESA & NASA/Solar Orbiter/EUI team, 2022 at Public Domain from Wiki Commons. Data processing by E. Kraaikamp. Everest: Wikivoyage banner for Mount Everest or Nepal by Fabien1309. This file is made available under the Creative Commons CC0 1.0 Universal Public Domain Dedication. Universe: The Observable Universe by Pablo Carlos Budassi from Wikipedia under Attribution-ShareAlike 3.0 Unported (CC BY-SA 3.0) in Public Domain.

ordered to paramagnetic phases, occurring for systems only a few nanometres in size[23,24]. The crossover temperature in a finite-size system hence can be described as an inflection point in $\langle|\mathbf{m}|\rangle$. The precise definition of a phase transition is significant when considering the main conclusions of Mermin and Wagner[5], which explicitly only apply in the case of an infinite system. As our results clearly show, sample sizes measured experimentally are not classifiable as infinite and, therefore, not subject to the Mermin-Wagner theorem. It is noteworthy that 3D compounds have weak dependence of their critical temperature on magnetic anisotropy[43]. Similar analysis performed for a finite 3D bulk system (Supplementary Fig. 8a, b) show that the inclusion of

anisotropy barely changes the results for $T_c$. This suggests that magnetism is an exchange-driven effect in both two and three dimensions.

On the practical side, heterostructures with conventional metallic magnetic materials could establish preferential directions of the magnetisation through anisotropic exchange and dipolar couplings. However, it is important to point out that the short-range order is enforced by the isotropic exchange couplings, and even a low anisotropy may suffice for stabilizing the direction of the magnetisation in the vdW layers, i.e., from underlying magnetic substrates. We can imagine micrometre-sized samples where all spins are still correlated at finite temperatures so it could represent a single bit. However, for miniaturization purposes multiple nanometre-sized bits are required on the same sample in order to be implemented in recording media. This is typically achieved by magnetic domains, but there are no domains in an isotropic model since the domain wall width is infinite. However, if vdW layers can be grown with grain boundaries, like in 2D mosaics[44], which are large enough that each grain area would have a uniform magnetisation, then a magnetic monolayer would have as many bits as available on the material surface. The underlying substrate hence would set the magnetisation direction for further implementations. This spin-interface engineering would be a considerable step towards on-demand magnetic properties at the atomic level given the flexibility on the orientation of the magnetic moments without a predefined direction at the layer. While the anisotropy circumvents the Mermin-Wagner theorem and causes the critical temperature $T_c$ to be nonzero in infinitely large systems, in finite samples the short-range order persists up to much higher temperatures ($T_x > T_c$) since $T_x$ is proportional to the isotropic exchange rather than the anisotropy[45,46]. Indeed, the long tail features observed in the intrinsic magnetisation (Fig. 2) extending above the crossover temperature suggest that short-range order is present. In addition, the existence of short-range order in bulk magnetic systems near and above the Curie temperature has been experimentally and theoretically discovered in elemental transition metals[47–49]. These studies indicate the persistence of magnetic ordering within the supposedly disordered phase above the Curie temperature, where any ordered phase is primarily controlled by exchange interactions as in the case for 2D magnets. For instance, in bcc-Fe a short-range order within 5.4 Å was found[47] which is much smaller than the magnitudes obtained in our simulations for vdW materials.

In conclusion, we presented large-scale spin dynamics simulations and analytical calculations on the temperature dependence of the intrinsic magnetisation in 2D magnetic materials described by an isotropic Heisenberg model. We found that short-range magnetic order at non-zero temperature is a robust feature of isotropic 2D magnets even at experimentally accessible length and time scales. Our data show that the often asserted Mermin-Wagner limit[5] does not apply to 2D materials on real laboratory sample sizes . Since the spins are aligned due to the exchange interactions already in the isotropic model, the direction of the magnetisation may be stabilized by geometrical factors or finite-size effects. These findings open up possibilities for a wider range of 2D magnetic materials in device applications than previously envisioned. Furthermore, the limited applicability of the analytical Mermin–Wagner theorem opens similar possibilities in other fields such as superconductivity[9] and liquid crystal systems[50], where the relevant length scale of correlations is known to be much greater than that required for experimental measurements and applications. Our results suggest that if the magnetic anisotropy can be controlled to a certain degree[51] until it completely vanishes, new effects of strongly correlated spins or more unusual disordered states may be observed.

## Methods
We used atomistic simulations methods[6,27,52–56] implemented in the VAMPIRE software[57] to compute the magnetic properties of 2D

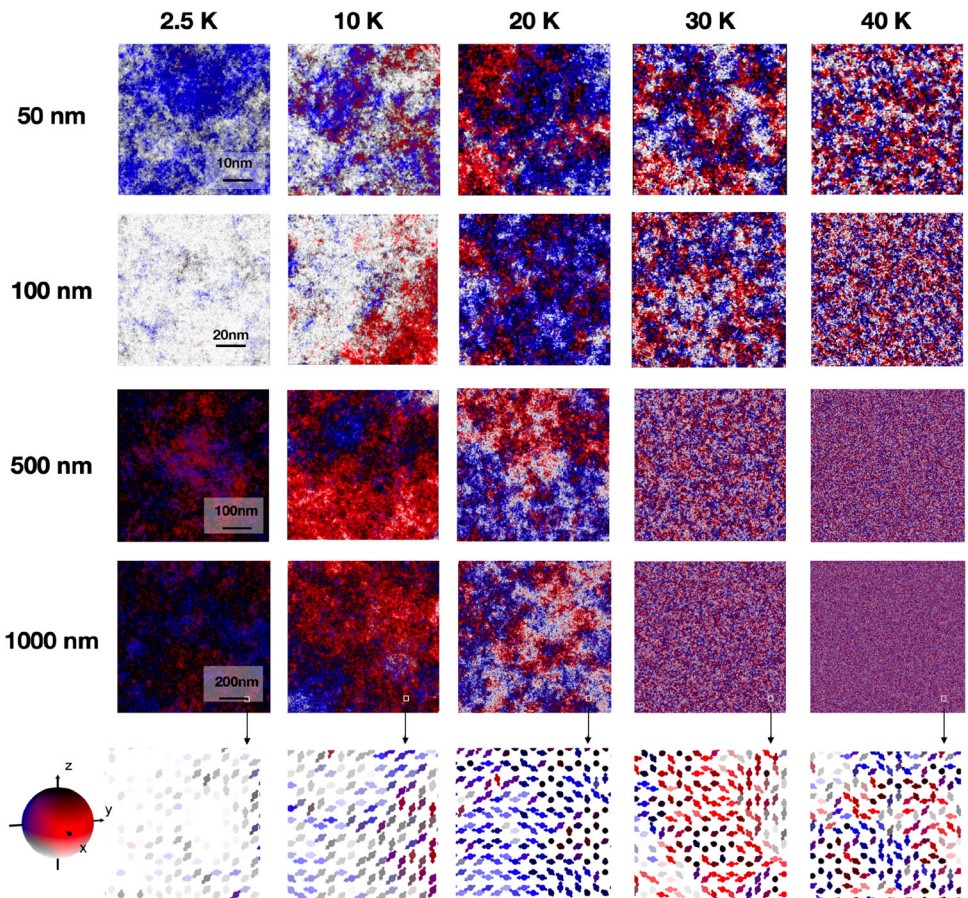

**Fig. 4 | Temperature-dependent magnetic order.** Visualisations of the magnetic spin configurations for the honeycomb lattice starting from an ordered state as a function of system size (vertical column) at different temperatures. The spins are projected following the colour scale shown in the sphere on the left. The bottom row shows a local view of the spins inside a 5 nm × 5 nm area at the location outlined by the small boxes in the 1000 × 1000 nm² snapshots.

magnetic materials. The energy of our system is calculated using the spin Hamiltonian:

$$\mathcal{H} = -\sum_{i<j} J_{ij}\mathbf{S}_i \cdot \mathbf{S}_j - K\sum_i \left(S_i^z\right)^2, \tag{4}$$

where $\mathbf{S}_{i,j}$ are unit vectors describing the local spin directions on magnetic sites $i, j$ and $J_{ij}$ is the exchange constant between spins. An easy-axis magnetocrystalline anisotropy constant $K$ can be included as well, with negligible modifications of the results as described in the text. Simulations were run for system sizes of 50 nm, 100 nm, 500 nm and 1000 nm laterally along the $x$ and $y$ directions with periodic boundary conditions (PBCs), and 1 atomic layer thick along the $z$ direction. Similar PBCs were used in the analytical model. However, simulation results using open boundary conditions (OBCs) ended up in similar conclusions (Supplementary Fig. 5). For the honeycomb lattice, the simulations were initialised in either a perfectly ordered state aligned along the $z$ direction or a random state corresponding to infinite temperature. For these simulations the final $\langle|\mathbf{m}|\rangle$ $(T)$ curves were identical to each other. However, at low temperatures it took ten times as many steps to reach the final equilibrium state from the random state, so for the remaining structures only simulations starting from the ordered states were run. The systems were integrated using a Monte Carlo integrator using a uniform sampling algorithm[57] to remove any bias introduced from more advanced algorithms[31]. To investigate the temperature dependence, the simulation temperature was varied from 0 to 90 K in 2.5 K steps. $40 \times 10^6$ Monte Carlo steps were run for each temperature step. This was split into $39 \times 10^6$ equilibration steps and then $10^6$ time steps from which the statistics

were calculated. The Monte Carlo simulations use a pseudo-random number sequence generated by the Mersenne Twister algorithm[58] due to its high quality, avoiding correlations in the generated random numbers and with an exceptionally long period of $2^{19937} - 1 \sim 10^{6000}$. The parallel implementation generates different random seeds on each processor to ensure no correlation between the generated random numbers.

The time-dependent simulations in Fig. 1c, d were performed by solving the stochastic Landau–Lifshitz–Gilbert equation:

$$\frac{\partial \mathbf{S}_i}{\partial t} = -\frac{\gamma_e}{1+\lambda^2}\left[\mathbf{S}_i \times \mathbf{B}_{\text{eff}} + \lambda\mathbf{S}_i \times \left(\mathbf{S}_i \times \mathbf{B}_{\text{eff}}\right)\right], \tag{5}$$

which models the interaction of an atomic spin moment $\mathbf{S}_i$ with an effective magnetic field $\mathbf{B}_{\text{eff}} = -1/\mu_s\, \partial\mathcal{H}/\partial\mathbf{S}_i$. The effective field causes the atomic moments to precess around the field, where the frequency of precession is determined by the gyromagnetic ratio of an electron ($\gamma_e = 1.76 \times 10^{11}$ rad s⁻¹T⁻¹) and $\lambda = 1$ is the damping constant. The large value of $\lambda$ was used to accelerate the relaxation dynamics in order to be computationally achievable (~72 hours). For a different damping, one has to wait longer or shorter for this to happen. Based on the system sizes used in our computations, this can vary between ~5 days up to several weeks, which is not practical. However, once the system is at equilibrium, the value of the damping is not important. Moreover, a large damping would correspond to large fluctuations on the magnitude of the magnetisation and its direction. Lower damping would lead to naturally slower dynamics of the magnetisation. Nevertheless, we barely noticed any at the timescale included in our work (Fig. 1c–d). It is worth mentioning that no

damping parameter are used in the Monte Carlo calculations which support our conclusions. The effect of temperature is taken into account using Langevin dynamics[59] (as in Eq. (5)), where the thermal fluctuations are represented by a Gaussian white noise term. At each time step the instantaneous thermal field acting on each spin is given by

$$\mathbf{B}^i_{\text{th}} = \sqrt{\frac{2\lambda k_B T}{\gamma \mu_s \Delta t}} \boldsymbol{\Gamma}(t) \qquad (6)$$

where $k_B$ is the Boltzmann constant, $T$ is the system temperature and $\boldsymbol{\Gamma}(t)$ is a vector of standard (mean 0, variance 1) normal variables which are independent in components and in time. The thermal field is added to the effective field in order to simulate a heat bath. The system was integrated using a Heun numerical scheme[57].

## Data availability
The data that support the findings of this study are available within the paper and its Supplementary Information.

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

## Acknowledgements

We thank David Mermin, Mikhail Katsnelson, and Bertrand Halperin for valuable discussions. L.R. gratefully acknowledges funding by the National Research, Development and Innovation Office of Hungary via Project No. K131938 and by the Young Scholar Fund at the University of Konstanz. U.A. gratefully acknowledges funding by the Deutsche Forschungsgemeinschaft (DFG, German Research Foundation)-Project-ID 328545488-TRR 227, Project No. A08; and grants PID2021-122980OB-C55 and RYC-2020-030605-I funded by MCIN/AEI/10.13039/501100011033 and by "ERDF A way of making Europe" and "ESF Investing in your future". E.J.G.S. acknowledges computational resources through CIRRUS Tier-2 HPC Service (ec131 Cirrus Project) at EPCC funded by the University of Edinburgh and EPSRC (EP/P020267/1); ARCHER UK National Supercomputing Service (http://www.archer.ac.uk) *via* Project d429. E.J.G.S. acknowledges the Spanish Ministry of Science's grant program "Europa-Excelencia" under grant number EUR2020-112238, the EPSRC Early Career Fellowship (EP/T021578/1), and the University of Edinburgh for funding support. K.S.N. is supported by the Ministry of Education, Singapore, under its Research Centre of Excellence award to the Institute for Functional Intelligent Materials (I-FIM, project No. EDUNC-33-18-279-V12) and by the Royal Society (UK, grant number RSRP\R\190000). For the purpose of open access, the authors have applied a Creative Commons Attribution (CC BY) licence to any Author Accepted Manuscript version arising from this submission.

## Author contributions

E.J.G.S. conceived the idea and supervised the project. S.J. performed the atomistic simulations with inputs from E.J.G.S. and R.F.L.E. L.R. and U.A. developed the semi-analytical model and undertook the numerical simulations. E.J.G.S. wrote the paper with a draft initially prepared by S.J. and R.F.L.E. and also with inputs from K.S.N., U.A. and L.R. All authors contributed to this work, read the manuscript, discussed the results, and agreed on the included contents.

## Competing interests

The authors declare no competing interests.
