## [Peer Review File · Nature Communications]

Reviewers' Comments:

Reviewer #1:

Remarks to the Author:

Comments to the author,

1. In the manuscript entitled "Breaking through the Mermin-Wagner limit in 2D van der Waals magnets" by Sarah Jenkins et al., the authors reported the breakthrough of Mermin-Wagner limit in 2D van der Waals magnets by large-scale spin dynamics simulations. The results are very interesting and should contribute to the exploration of 2D magnets. This manuscript is suggested to be published in Nature Communications before the authors addressed the following comments. The size effects have been thoroughly investigated by spin dynamic simulation, so, does the boundary effect should be considered in the finite size simulation?
2. As it known, the prerequisite of Mermin-Wagner theorem is isotropic Heisenberg model and short-range exchange interaction. In this manuscript, the authors considered the effects of both magnetic anisotropy energy and the cell size on cross temperature, and concluded that only short-range order defined by the isotropic interactions dominates. Can the authors consider the influence of the degree, orientation of short-range order and interaction strength on the cross-temperature to further support the point of view in the manuscript?
3. It is generally believed that a large magnetic anisotropy is needed to stabilize the magnetic order, while according to the authors' point in the manuscript, the transition temperature has little to do with the magnetic anisotropy. Then, according to what rule can one find the low-dimensional magnetic materials with high transition temperature?
4. The studies of 2D van der Waals magnets by computational methods have been thoroughly reviewed in Wiley Interdisciplinary Reviews: Computational Molecular Science, 12(2), e1545, which is suggested to be considered in this manuscript.
5. There are some format errors that should be double-checked again, for example, the sentence "Jij is the exchange constant between spins. An easy-axis magneto-crystalline anisotropy constant K" in page 14 misses a space between "spin." and "An".

Reviewer #2:

Remarks to the Author:

This manuscript reports results of detailed Monte-Carlo and micromagnetic simulations of the classical isotropic nearest-neighbor Heisenberg model on finite 2D lattices. The sizes of the systems considered in the manuscript are compatible with those of currently available samples of so-called "flakes." The authors find that short-range order, characterized by a space-integrated correlation function (the author's intrinsic magnetization, eq. 2) is present at non-zero temperatures. They use a Landau-Lifshitz-Gilbert equation to investigate the time evolution of the short-range order at finite temperatures and conclude that the magnetization direction is stable over times long enough to be relevant in practice. They determine crossover temperatures for the vanishing of short-range order that are compatible with the ordering temperature found in recent experiments on van der Waals ferromagnets. They conclude that the main driving force behind the existence of short-range order in the Heisenberg model is not anisotropy, as frequently quoted in the literature, but finite-size effects.

This manuscript presents a relevant and useful numerical study of the limitations of the Hohenberg-Mermin-Wagner (HMW) theorem for real samples of finite size. The authors are right to point out (lines 49-53 of the manuscript) that, embedded in the requisites of the theorem, lies the reason for their findings: the theorem requires that the thermodynamic limit is taken. They also rightly point out that there is a widespread belief in the scientific community that the sizes of real samples are usually enough to attain this limit in practice. However, systematic discussions of the actual values needed to recover the conditions imposed by the theorem are surprisingly scarce in the literature. The results presented here are relevant contributions to this discussion.

Although scarce, there are examples of this discussion in the literature that the authors have missed. For instance, Using a combination of analytical methods and Monte Carlo simulations, Kapikranian et al. [J. Phys. A: Math. Theor. 40 (2007) 3741] have shown that the isotropic Heisenberg model on a finite lattice displays short-range order, characterized by a power-law

decay of the spin-spin correlation function. This is in line with the findings reported in this manuscript. More recently, Palle and Sunko [J. Phys. A: Math. Theor. 54 (2021) 315001], have provided a very readable discussion. Their references 16-23 are part of the history of this discussion and also relevant for the present manuscript. Reference 9, in particular, provides an illuminating qualitative discussion of this subject, and even an amusing comparison, that is reminiscent of the ones used by the authors in Fig. 3 (the following sentences can be found on the bottom of page 9 of <https://courses.physics.illinois.edu/phys598PTD/fa2013/L9.pdf>): "e.g. in the case of crystalline order T_0 is replaced by a temperature that is at least of the order of the Debye temperature, so to see HMW-type effects in (say) a graphene crystal at a few degrees would require the crystal to extend from here to the moon! The moral is that before taking the theorem too seriously in a real-life situation, one should carefully put in the numbers."

Another discussion with a long history is that of the existence of short-range order in bulk magnetic systems above the Curie temperature. Experimental evidence in its favor has been found in elemental transition metals [PRL 54, 932 (1985), PRL 48, 1686 (1982)]. In PRB 72, 140406(R) (2005), Antropov showed, using TDDFT, that local moments in Ni and Fe above T_c display what he terms "giant magnetic short-range order." Although those results may seem weakly related to the present discussion, I think the relationship is stronger than it seems. They indicate the persistence of order well within the supposedly "disordered" phase of systems in which order is driven essentially by exchange, one of the main points made by the present manuscript.

Finally, two technical issues:

- The sentence "The magnitude of J_{ij}/k_B is within the same range as those observed for CGT (with a critical temperature of 66 K)² where a negligible magnetic anisotropy ($< 1 \mu\text{eV}$) was observed for thin layers but a stable magnetic signal was still measured at finite temperatures ($\sim 4.7 \text{ K}$)," found on page 5, is confusing. According to ref. 2, T_c for the monolayer is negligible, in line with the extremely low anisotropy value they infer. The much higher $T_c=66\text{K}$ quoted in this manuscript is actually a property of thick samples, as stated in the beginning of the "Methods" section of Ref. 2.
- The value of the damping constant the authors used in their LLG equation seems exaggerated for a system described by the isotropic Heisenberg model. In insulating magnets with negligible spin-orbit coupling and no disorder, the Gilbert damping should also be negligible. The large value used here may have implications for the results presented in Fig. 1C-d and Fig. 4, since in the presence of a random thermal field the amplitude of low-energy modes in the stationary state could be substantially modified by the damping.

In conclusion, I believe the the manuscript can be published in Nature Communications after the issues listed above have been addressed.

Reviewer #3:

Remarks to the Author:

In 1966, Mermin and Wagner showed a rigorous proof, that there can be neither ferro- nor antiferromagnetic ordering for the one- or two-dimensional systems described by isotropic Heisenberg models with finite-range interactions, at non-zero temperatures. The magnetic order found in low-dimensional systems has, since then, usually been viewed as an effect of the magnetic anisotropy of the system. It is important to note that the Mermin-Wagner theorem assumes systems of infinite lateral dimensions. When studying the microscopic phenomena, for many purposes, the samples of novel 2D materials produced in the labs nowadays can indeed be considered infinite. However, one has to be aware that they are in fact finite. Therefore, when considering the Mermin-Wagner theorem, one has to ask themselves just how infinite is "infinite" and at which lateral size the sample can be viewed to be outside of the validity space of the theorem. The Authors of the current manuscript have tackled this problem numerically with the aid of atomistic simulations and analytical models. The result is, at least for me, quite surprising: even for the 2D-systems with the sizes comparable to the diameter of the observable universe, magnetic ordering at finite temperature is possible! Importantly, it is shown that this result is independent of the magnetic anisotropy and it is actually the exchange interactions that are

responsible for the effect.

The methods applied in this work are appropriate and well-described, the calculations seem valid and carefully performed. Relevant research is properly cited. The manuscript is well-written and timely, boosting the importance of 2D-materials for applications in data-storage technologies. It sheds light on the confusing presence of magnetism in experimental samples, even though the confusion is mainly rooted in our misconception of the region of validity of Mermin-Wagner theorem. I am glad this issue seems to be settled now and recommend the work for publication, in its current form.

REVIEWER COMMENTS

Reviewer #1 (Remarks to the Author):

Comments to the author,

1. In the manuscript entitled “Breaking through the Mermin-Wagner limit in 2D van der Waals magnets” by Sarah Jenkins et al., the authors reported the breakthrough of Mermin-Wagner limit in 2D van der Waals magnets by large-scale spin dynamics simulations. The results are very interesting and should contribute to the exploration of 2D magnets. This manuscript is suggested to be published in Nature Communications before the authors addressed the following comments.

Response: *We thank the Reviewer for the kind comments about our manuscript, and for considering our work interesting and suggesting publication in Nature Communications. We have addressed below all points raised by the Reviewer and modify the manuscript accordingly.*

The size effects have been thoroughly investigated by spin dynamic simulation, so, does the boundary effect should be considered in the finite size simulation?

Response: *We have checked that our conclusions do not change by using different boundary conditions, such as using open boundary conditions (OBCs) as included in Supplementary Fig. 5, which is different than the periodic boundary conditions (PBCs) used in the paper. We have also tested that finite sizes with a different shape (e.g. circular) resulted in similar results. Therefore, boundaries/edges, and shapes are not observed to provide any modification on the conclusions already mentioned in the manuscript.*

We have included additional discussions in the text (page 8, lines 140-151; page 15, lines 280-284) regarding the PBCs and the circular shape, with Supplementary Figures 5-6 supporting our discussions.

2. As it known, the prerequisite of Mermin-Wagner theorem is isotropic Heisenberg model and short-range exchange interaction. In this manuscript, the authors considered the effects of both magnetic anisotropy energy and the cell size on cross temperature, and concluded that only short-range order defined by the isotropic interactions dominates. Can the authors consider the influence of the degree, orientation of short-range order and interaction strength on the cross-temperature to further support the point of view in the manuscript?

Response: *In the isotropic Heisenberg model used in our simulations and in the Mermin-Wagner theorem the short-range interactions are isotropic to any direction confined in the unit sphere. Hence, we cannot control their orientation as the spins are not fixed. Indeed, the intrinsic magnetisation (Eq. 2) is a scalar and not a vector quantity which rules out any preferential direction. However, we have quantified how the spins are oriented relative to the mean magnetisation direction at different temperatures and flake sizes as included in Supplementary Figure 4 and discussed in the text. We also include the spin-spin correlation (thus the average degree of alignment as a function of distance) to show the relative orientation of the short-range order in Supplementary Figure 7.*

Regarding the interaction strength, our tests showed that different magnitudes of the exchange constants would result in a linear re-scaling of the crossover temperatures which do not affect the conclusions.

We have included additional discussions in the main text (page 8, lines 140-151) and Supplementary Figures 5-6 explaining further the points mentioned by the Reviewers.

3. It is generally believed that a large magnetic anisotropy is needed to stabilize the magnetic order, while according to the authors' point in the manuscript, the transition temperature has little to do with the magnetic anisotropy. Then, according to what rule can one find the low-dimensional magnetic materials with high transition temperature?

Response: *As demonstrated in our manuscript, the exchange interactions are the key factor for the stabilisation of magnetism in 2D. This applies for either low or high transition temperature materials, and not on the value of the magnetic anisotropy itself. Then, if materials can be engineered to have high exchange coupling between atoms via different chemical synthesis, they may result in high temperature magnets.*

We have included additional discussions in the manuscript regarding this point raised by the Reviewer (page 8, lines 140-151).

4. The studies of 2D van der Waals magnets by computational methods have been thoroughly reviewed in Wiley Interdisciplinary Reviews: Computational Molecular Science, 12(2), e1545, which is suggested to be considered in this manuscript.

Response: *We thank the Reviewer for the reference. It has been included in the updated version of the manuscript.*

5. There are some format errors that should be double-checked again, for example, the sentence "Jij is the exchange constant between spins. An easy-axis magneto-crystalline anisotropy constant K" in page 14 misses a space between "spin." and "An".

Response: *We thank the Reviewer for pointing them out. We have amended them in the updated version of the manuscript.*

Reviewer #2 (Remarks to the Author):

This manuscript reports results of detailed Monte-Carlo and micromagnetic simulations of the classical isotropic nearest-neighbor Heisenberg model on finite 2D lattices. The sizes of the systems considered in the manuscript are compatible with those of currently available samples of so-called "flakes." The authors find that short-range order, characterized by a space-integrated correlation function (the author's intrinsic magnetization, eq. 2) is present at non-zero temperatures. They use a Landau-Lifshitz-Gilbert equation to investigate the time evolution of the short-range order at finite temperatures and conclude that the magnetization direction is stable over times long enough to be relevant in practice. They determine crossover temperatures for the vanishing of short-range order that are compatible with the ordering temperature found in recent experiments on van der Waals ferromagnets. They conclude that the main driving force behind the existence of short-range order in the Heisenberg model is not anisotropy, as frequently quoted in the literature, but finite-size effects.

This manuscript presents a relevant and useful numerical study of the limitations of the Hohenberg-Mermin-Wagner (HMW) theorem for real samples of finite size. The authors are right to point out (lines 49-53 of the manuscript) that, embedded in the requisites of the theorem, lies the reason for their findings: the theorem requires that the thermodynamic limit

is taken. They also rightly point out that there is a widespread belief in the scientific community that the sizes of real samples are usually enough to attain this limit in practice. However, systematic discussions of the actual values needed to recover the conditions imposed by the theorem are surprisingly scarce in the literature. The results presented here are relevant contributions to this discussion.

Response: *We thank the Reviewer for the kind words regarding our manuscript. We have addressed below all points raised by the Reviewer and modify the manuscript accordingly.*

Although scarce, there are examples of this discussion in the literature that the authors have missed. For instance, Using a combination of analytical methods and Monte Carlo simulations, Kapikranian et al. [J. Phys. A: Math. Theor. 40 (2007) 3741] have shown that the isotropic Heisenberg model on a finite lattice displays short-range order, characterized by a power-law decay of the spin-spin correlation function. This is in line with the findings reported in this manuscript. More recently, Palle and Sunko [J. Phys. A: Math. Theor. 54 (2021) 315001], have provided a very readable discussion. Their references 16-23 are part of the history of this discussion and also relevant for the present manuscript. Reference 9, in particular, provides an illuminating qualitative discussion of this subject, and even an amusing comparison, that is reminiscent of the ones used by the authors in Fig. 3 (the following sentences can be found on the bottom of page 9 of <https://courses.physics.illinois.edu/phys598PTD/fa2013/L9.pdf>): “e.g. in the case of crystalline order T_0 is replaced by a temperature that is at least of the order of the Debye temperature, so to see HMW-type effects in (say) a graphene crystal at a few degrees would require the crystal to extend from here to the moon! The moral is that before taking the theorem too seriously in a real-life situation, one should carefully put in the numbers.”

Response: *We have included all the references mentioned by the Reviewer as well as additional discussions in the manuscript regarding the historical aspects and estimations by A. J. Leggett on graphene samples (page 3, lines 53-56; and page 10, lines 183-186).*

Another discussion with a long history is that of the existence of short-range order in bulk magnetic systems above the Curie temperature. Experimental evidence in its favor has been found in elemental transition metals [PRL 54, 932 (1985), PRL 48, 1686 (1982)]. In PRB 72, 140406(R) (2005), Antropov showed, using TDDFT, that local moments in Ni and Fe above T_c display what he terms “giant magnetic short-range order.” Although those results may seem weakly related to the present discussion, I think the relationship is stronger than it seems. They indicate the persistence of order well within the supposedly “disordered” phase of systems in which order is driven essentially by exchange, one of the main points made by the present manuscript.

Response: *We thank the Reviewer for pointing these papers out. We have included additional discussions regarding the stabilisation of ordered phase near and above the Curie temperatures in the manuscript (pages 13-14, lines 253-261).*

Finally, two technical issues:

- The sentence “The magnitude of J_{ij}/k_B is within the same range as those observed for CGT (with a critical temperature of 66 K)² where a negligible magnetic anisotropy ($< 1 \mu\text{eV}$) was observed for thin layers but a stable magnetic signal was still measured at finite temperatures (~ 4.7 K),” found on page 5, is confusing. According to ref. 2, T_c for the monolayer is negligible, in line with the extremely low anisotropy value they infer. The much

higher $T_c=66\text{K}$ quoted in this manuscript is actually a property of thick samples, as stated in the beginning of the “Methods” section of Ref. 2.

Response: *We thank the reviewer for the comments. We have re-written those sentences (page 5, lines 90-96) and clarify the selection of the exchange interactions for our simulations.*

- The value of the damping constant the authors used in their LLG equation seems exaggerated for a system described by the isotropic Heisenberg model. In insulating magnets with negligible spin-orbit coupling and no disorder, the Gilbert damping should also be negligible. The large value used here may have implications for the results presented in Fig. 1C-d and Fig. 4, since in the presence of a random thermal field the amplitude of low-energy modes in the stationary state could be substantially modified by the damping.

Response: *The large value of the damping ($\lambda=1$) was used to accelerate the spin dynamics to reach the equilibrium at a computationally achievable time (~ 72 hours). For a different damping, one has to wait longer or shorter for this to happen. Based on the system sizes used in our computations this can vary between ~ 5 days up to several weeks, which is not computationally practical. However, once the system is at equilibrium, the value of the damping is not important as it is the case in our results. Moreover, a large damping would correspond to large fluctuations on the magnitude of the magnetization (Fig. 1c) and its direction (Fig. 1d) and so is the low time limit. Lower damping would lead to naturally slower dynamics of the magnetization. Nevertheless, we barely noticed any at the timescale included in our manuscript. It is worth mentioning that no damping parameter is present in the Monte Carlo calculations which support our conclusions.*

We have included additional comments in the manuscript highlighting the value of the damping used in our simulations (pages 16).

In conclusion, I believe the the manuscript can be published in Nature Communications after the issues listed above have been addressed.

Response: *We hope to have addressed all the comments by the Reviewer and our manuscript can be finally accepted in Nature Communications.*

Reviewer #3 (Remarks to the Author):

In 1966, Mermin and Wagner showed a rigorous proof, that there can be neither ferro- nor antiferromagnetic ordering for the one- or two-dimensional systems described by isotropic Heisenberg models with finite-range interactions, at non-zero temperatures. The magnetic order found in low-dimensional systems has, since then, usually been viewed as an effect of the magnetic anisotropy of the system. It is important to note that the Mermin-Wagner theorem assumes systems of infinite lateral dimensions. When studying the microscopic phenomena, for many purposes, the samples of novel 2D materials produced in the labs nowadays can indeed be considered infinite. However, one has to be aware that they are in fact finite. Therefore, when considering the Mermin-Wagner theorem, one has to ask themselves just how infinite is “infinite” and at which lateral size the sample can be viewed to be outside of the validity space of the theorem. The Authors of the current manuscript have tackled this problem numerically with the aid of atomistic simulations and analytical models. The result is, at least for me, quite surprising: even for the 2D-systems with the sizes comparable to the diameter of the observable universe, magnetic ordering at finite temperature is possible! Importantly, it is shown that this result is independent of the magnetic anisotropy and it is

actually the exchange interactions that are responsible for the effect.

The methods applied in this work are appropriate and well-described, the calculations seem valid and carefully performed. Relevant research is properly cited. The manuscript is well-written and timely, boosting the importance of 2D-materials for applications in data-storage technologies. It sheds light on the confusing presence of magnetism in experimental samples, even though the confusion is mainly rooted in our misconception of the region of validity of Mermin-Wagner theorem. I am glad this issue seems to be settled now and recommend the work for publication, in its current form.

Response: *We thank the Reviewer for the kind words regarding our manuscript, and for accepting it in its current form.*

Reviewers' Comments:

Reviewer #1:

Remarks to the Author:

Comments to the author,

All the comments raised by the reviewers have been properly replied and included in the manuscript. I recommend acceptance of the revised manuscript in Nature Communications.

Reviewer #2:

Remarks to the Author:

It is my opinion that the authors addressed all the points raised by me and the other referee satisfactorily. I recommend the publication of the manuscript in its present form.